# Day-Ahead Forecast of Electric Vehicle Charging Demand with Deep Neural Networks †

**Gilles Van Kriekinge \*** , **Cedric De Cauwer** , **Nikolaos Sapountzoglou** and **Thierry Coosemans** and **Maarten Messagie**

EVERGi Research Group, MOBI Research Centre & ETEC Department, Vrije Universiteit Brussel (VUB), Pleinlaan 2, 1050 Brussel, Belgium; Cedric.De.Cauwer@vub.be (C.D.C.); nikolaos.sapountzoglou@vub.be (N.S.); thierry.coosemans@vub.be (T.C.); maarten.messagie@vub.be (M.M.)

\* Correspondence: givkriek@vub.be

† This paper is an extended version of our paper published in 34th International Electric Vehicle Symposium and Exhibition (EVS34), Nanjing, China, 25–28 June 2021.

**Abstract:** The increasing penetration rate of electric vehicles, associated with a growing charging demand, could induce a negative impact on the electric grid, such as higher peak power demand. To support the electric grid, and to anticipate those peaks, a growing interest exists for forecasting the day-ahead charging demand of electric vehicles. This paper proposes the enhancement of a state-of-the-art deep neural network to forecast the day-ahead charging demand of electric vehicles with a time resolution of 15 min. In particular, new features have been added on the neural network in order to improve the forecasting. The forecaster is applied on an important use case of a local charging site of a hospital. The results show that the mean-absolute error (MAE) and root-mean-square error (RMSE) are respectively reduced by 28.8% and 19.22% thanks to the use of calendar and weather features. The main achievement of this research is the possibility to forecast a high stochastic aggregated EV charging demand on a day-ahead horizon with a MAE lower than 1 kW.

**Keywords:** aggregated charging demand; day-ahead forecast; electric vehicle; feature importance; recurrent neural network

## 1. Introduction

The Intergovernmental Panel on Climate Change (IPCC) has further confirmed that climate is warming up due to human activities which are the principal source of carbon dioxide emissions in the atmosphere [1]. To reduce such emissions, new technologies are being massively implemented such as wind energy, solar energy, and electric vehicles (EV). Electric vehicles are presented as a sustainable alternative to conventional internal combustion engine (ICE) vehicles. In a Sustainable Development Scenario (SDS) 2020–2030, the EV share is expected to grow exponentially up to 13.4% in 2030 [2]. While EVs have many advantages compared to ICE vehicles, such an increase in number of electric vehicles will have negative consequences on the electric grid by inducing higher peak powers, frequency and voltage deviations and an overall increase in energy demand [3].

In order to mitigate these problems, electric vehicles can be charged intelligently by spreading or shifting the charging demand over time, according to user and electricity system needs. This coordination of EV charging provides a complex optimization problem that could benefit from EV charging demand forecast. This research focuses on developing accurate EV charging demand forecasters, which can be used by energy management systems for coordinated EV charging.

This paper is organized as follows. Section 2 gives an introduction to the methods and benefits of coordinated smart charging, provides an overview of the relevant literature on the EV charging demand forecasting of small and large EV fleets, and highlights the contribution of this research with respect to the literature gaps. Section 3 provides a

complete overview of the neural networks configurations including data pre-processing, neural network detailed characteristics, and forecast post-processing. Then, Section 4 shows the specificity of the use case under study and the simulations results, including the loss function evaluation, forecast examples, and a feature importance analysis. Finally, Section 5 concludes the results of this research.

## 2. Literature Review

For the management of coordinated smart charging, literature has proposed many different methods, of which a couple of them are discussed here. Authors in [4] propose a two-stage optimization scheduler which can both satisfy the EV charging demand needs and take advantage of low-load periods. A similar objective is studied in [5] but including bi-directional charging. The results show the possibility to reduce the emissions and the investments in peak load plants. In [6], a bi-directional scheduling algorithm is developed as distributed frequency regulation source and shows the great opportunity for Vehicle-to-Grid (V2G) to take part as frequency regulator. Using bi-directional charging, it is also possible to facilitate the integration of renewable energy sources (RES), as shown in [7]. Finally, a review is available in [8] which summarizes different types of control algorithms to schedule the charging of electric vehicles in a smart grid context.

These methods for coordinated smart charging could help local energy system (LES) operators (e.g., microgrid operators or EV aggregators) and grid operators (transmission system operators (TSO) and distributions system operators (DSO)), to better manage the electricity grid, by anticipating the charging demand of EVs. For grid operators, short-term forecast of the charging demand of large EV fleets could help better dispatch generation units, enhance the safety of electric grid, reduce congestion problems, help regulating the grid stability, etc. [9,10]. For LES operators, the energy management systems (EMS) could include short-term forecast of the EV charging demand in order to enhance the operational optimization of its assets by using predictive control optimization techniques (for instance model predictive control algorithms) [11].

Several scientific studies offer an analysis of the forecast of the aggregated charging demand of large EV fleets on a short-term horizon. Authors in [12] propose a Seasonal AutoRegressive Integrated Moving Average with eXogenous regressors (SARIMAX) and compare it to a persistence benchmark. The results show improved performances by more than 26%. In [13], through a hierarchical approach where the forecasting problem is decomposed to sub-problems, the authors managed to improve their forecasting accuracy by 9.5%. Authors in [14], study different forecast algorithms from traditional statistical algorithms to artificial intelligence algorithms. The study shows that the neural network algorithm performed best, mainly on peak demand prediction. Finally, reference [10] proposes an AutoRegressive Integrated Moving Average (ARIMA) forecaster improved by tuning its parameters in order to minimize the mean-square error (MSE). In addition, it uses the EV charging demand forecast in a scheduling problem and it shows better unit commitment, as well as a reduction in operating cost.

Nonetheless, few papers forecast smaller EV fleets of typical charging sites such as office buildings, hotels, or shops with higher EV drivers stochastic behavior. Three papers [15–17], from the same authors, forecast the EV charging demand of a campus with 15 to 28 outlets with different algorithms, such as Modified Pattern-based Sequence Forecasting (MPSF), ARIMA, Support Vector Regression (SVR), Random Forest (RF), etc. Their specificity lies in a trade-off between computational time and forecast accuracy, which is important for their use case. The results show that the best algorithm to use in forecasting is the MPSF.

A well-known type of neural network called recurrent neural network (RNN) is more and more used in literature for forecast applications. For instance, such type of neural network can be used for wind power forecast [18], multinational trade forecast [19], peak wave energy period forecast [20], natural gas demand forecast [21], residential electricity load forecast [22], and electricity prices forecast [23].

Among such examples, two recent studies, References [24,25], from the same authors, have implemented this algorithm to forecast the EV charging demand. The simulation results show the superior performance of Long Short-Term Memory (LSTM) network, which is a kind of RNN, compared to conventional timeseries forecasters. However, the two previous papers study either the use case of large EV fleets or forecast on a super-short-term horizon (minute level). In [26], authors propose to forecast the aggregated EV charging demand for the next hour (using previous 24 h demand) by developing an Ensemble Learning approach combining an artificial neural network (ANN), an RNN, and an LSTM. The results show better performances than the individual neural networks.

*Research Gap and Contribution*

This literature review shows the lack of research on forecasting the EV charging demand. This is especially true on small EV fleets which tend to be more difficult to forecast, due to a higher stochastic behavior, short-term horizon, and high time resolution. Consequently, this paper builds on previous works by forecasting a small EV fleet on a day-ahead horizon and on a 15 min timestep resolution. The small EV fleet dataset is based on real data of a hospital semi-public charging site. To achieve better forecast results of such a difficult use case, multiple new contributions are included such as additional input features for the neural network, a variable learning rate function, and a post-processing of the forecast. These new contributions are compared with a state-of-the-art forecaster. Finally, an analysis is performed on the importance of individual features used to train the neural networks.

## 3. Materials and Methods

A scheme of the framework is shown in Figure 1 where each subsection's name and number are given.

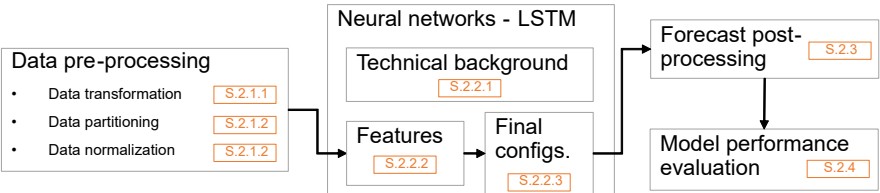

**Figure 1.** Framework of the models.

Firstly, a data pre-processing step, explained in Section 3.1, is needed in order to transform, divide, and normalize the data. Then, Section 3.2 focuses on the neural networks used to forecast. It includes a technical background, an in-depth analysis on the input features, and a summary on the different neural networks' configurations. Finally, a post-processing of the forecast in Section 3.3 and a model performance evaluation in Section 3.4 are provided.

### 3.1. Data Pre-Processing

#### 3.1.1. Data Transformation

The raw data contain individual EV charging sessions data which consist of: (a) the EV users' radio frequency identification (RFID), (b) the arrival and departure times, and (c) the energy consumed (in kWh). Such raw individual EV charging session data have to be transformed into an EV charging demand profile (in kW) in order to comply with the objective of this paper which is to forecast the aggregated EV charging demand on a 15 min timestep. In other words, each individual EV charging session from the raw data needs to be transformed into an individual EV charging profile, and each individual EV charging profile is stacked up to obtain a final aggregated EV charging demand timeseries.

To transform the raw individual EV charging session data into an individual EV charging profile, a common method is to consider the actual charging power (uncoordi-

nated charging) used to express it as a power over time, hence a timeseries. However, on an EMS perspective, uncoordinated charging power forecast is only useful when the chargers are non-controllable. In the case of smart or bi-directional chargers, a different forecast is needed that includes somehow the energy needs and the user flexibility. In addition, the charging power value of the charging sessions is not available, hence it is not possible to build an uncoordinated EV charging power profile. These are the reasons why a different method is used to obtain an individual EV charging profile. The latter consists of computing an average power, $P_{avg}$, using (1) similarly to [17].

$$P_{avg} = \frac{\text{Energy consumed [kWh]}}{\text{Parking time [h]}} \tag{1}$$

For example, a charging session of 10 kWh over a parking time of 10 h gives an average power of 1 kW over the whole charging session. The advantage of this method lies in the fact that it indirectly includes the charging flexibility of the EV users. For instance, with the same previous example but with a lower flexibility of 1 h parking time, the average power will be equal to 10 kW instead of 1 kW. Finally, Figure 2 summarizes graphically the method used in this paper to obtain an aggregated EV charging demand.

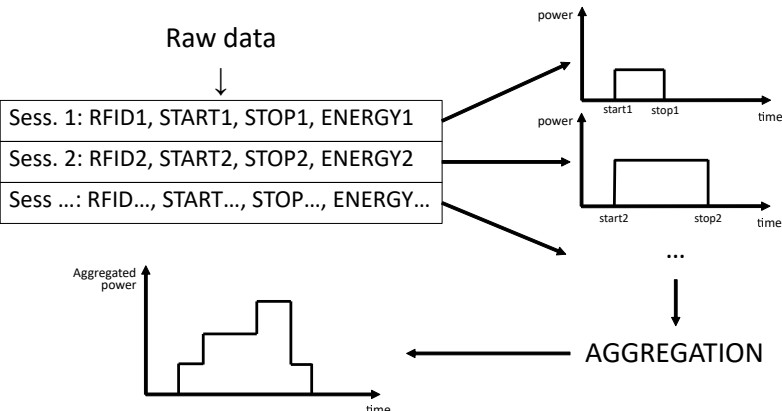

**Figure 2.** From raw data to aggregated EV charging power profile.

### 3.1.2. Data Partitioning and Normalization

A first and important step is to divide the timeseries into three subsets, mainly a training, a validation, and a testing subset, following the standard practice of a 0.7/0.2/0.1 partitioning. After that, the training subset (including all features explained in Section 3.2.2) is normalized before entering the neural network. Two different normalization techniques have been tested, called *z-score* and *min-max* normalization. The min-max normalization, after a preliminary test, showed clearly better performances, similarly to what has been observed in literature, and was thus the one selected for this paper. The formula of the min-max normalization is given by (2).

$$x_N = \frac{x_t - x_{min}}{x_{max} - x_{min}} \tag{2}$$

where $x_t$ is the non-normalized data point, $x_{min}$ the minimum value, and $x_{max}$ the maximum value of all data points.

### 3.2. Neural Networks

### 3.2.1. Technical Background

An artificial neural network (ANN) is a network made of multiple layers containing neurons interconnected between them. The interconnections are made of weights which have to be defined. There are many different classes of ANN which are used in many different applications. A particular class of ANN is of interest for forecast applications which is called Recurrent Neural Network (RNN). The specificity lies in the sequential

temporal dimension where previously learned information is used to define the next weights. By remembering the past, they make better decisions for the future, which is what is needed to forecast timeseries. The difference between a classical ANN and a RNN is represented in blue in Figure 3. The blue arrows show that at an instant $t$, the RNN receives information at time $t$ but also information previously learned at time $t-1$. Mathematically, such new links are given by (3) and (4) [26].

$$h_t = f(Uh_{t-1} + Wx_t + b) \tag{3}$$

$$y_t = g(Vh_t + c) \tag{4}$$

where $h_t$ and $h_{t-1}$ are the hidden layers state at time $t$ and $t-1$, $U$, $V$, and $W$ are the weight matrices, $b$ and $c$ are the biases, and $f$ and $g$ are the activation functions.

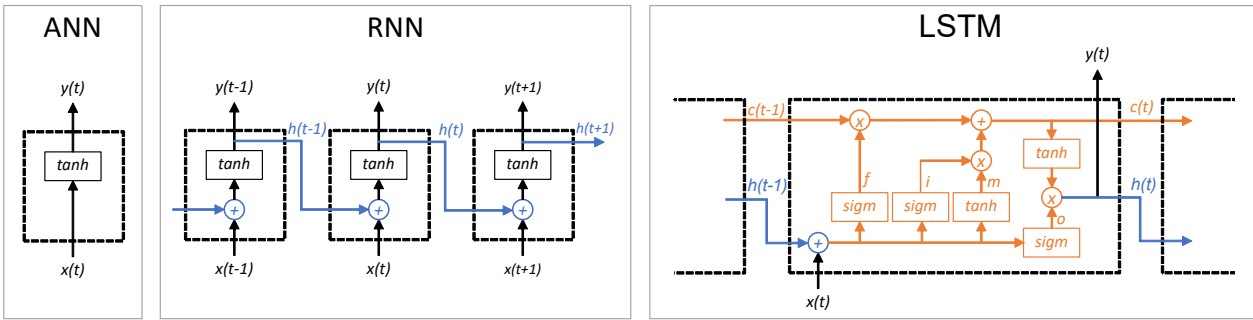

**Figure 3.** Different simplified representations of neurons of an ANN, an RNN, and an LSTM.

There exist different RNN architectures, and among them, the most important one is the long short-term memory (LSTM) network [27]. This is the type of neural network used to forecast the EV charging demand in this paper. They solve one of the biggest problems of RNN which is that such neural networks do not learn from long term information. They have what is called a vanishing gradient problem [28]. Such problem is solved in LSTM networks by adding specific gating mechanisms in the recurrent feedback loop. The gating mechanism controls what information should be kept or discarded while the neural network learns. The difference between a classical RNN and an LSTM network is represented in orange in Figure 3. More precisely, three new gates and one new unit are added in the LSTM neurons; an input gate $i$, a forget gate $f$, an output gate $o$, and an internal memory unit $m$. Their specific representations are given mathematically by (5)–(10) and graphically in Figure 3 at time $t$.

$$i_t = \sigma(W_i(x_t + h_{t-1}) + b) \tag{5}$$

$$f_t = \sigma(W_f(x_t + h_{t-1}) + b) \tag{6}$$

$$o_t = \sigma(W_o(x_t + h_{t-1}) + b) \tag{7}$$

$$m_t = tanh(W_m(x_t + h_{t-1}) + b) \tag{8}$$

$$c_t = f_t \times c_{t-1} + i_t \times m_t \tag{9}$$

$$h_t = o_t \times tanh(c_t) \tag{10}$$

where $W_{i,f,o,m}$ are the weight matrices associated to the corresponding gates and unit. For additional information on ANN, RNN, and LSTM networks, readers are referred to [27]. Multiple LSTM configurations are tested in this paper and compared with the configuration from [25], denoted LSTM-B. All configurations contain an input layer, two hidden layers, and an output layer. It has been decided to use two hidden layers because they give the ability to obtain a deep learning configuration and thus, extract higher level information

from the timeseries [24]. The neural networks are implemented in Python using *Tensorflow* platform and *Keras* library [29].

### 3.2.2. Features

Three different neural networks are tested in this paper denoted as LSTM-B, LSTM-C, and LSTM-W. The first neural network, called LSTM-B, is used as reference, where 'B' refers to 'Base'. The second neural network, called LSTM-C, includes additional calendar features, where 'C' refers to 'Calendar'. The last neural network, called LSTM-W, includes additional weather features, where 'W' refers to 'Weather'. The calendar features are included because the end-user behavior depends on the type of day considered (working day or weekend, holiday or not, and the day of the week) and the time of the day. The weather features tested in this paper are the daily outside temperature and daily rainfall, because the individual EV energy demand is expected to change in function of the outside temperature and rainfall as shown in [30–32]. The list of features, partially based on [24,33], is given in Table 1.

**Table 1.** Input layer features.

| Class | Feature | LSTM-B | LSTM-C | LSTM-W |
|---|---|---|---|---|
| Load | EV charging demand [kW] | X | X | X |
| | Average weekly EV demand [kW] | | X | X |
| Calendar | Quarter-hour number [/] | | X | X |
| | Day number [/] | | X | X |
| | Binary working day [0 or 1] | | X | X |
| | Binary Holiday [0 or 1] | | X | X |
| Weather | Daily temperature [°C] | | | X |
| | Daily rainfall [mm/h] | | | X |

Two of the calendar features, quarter-hours (denoted $\Delta T$) and day number (denoted $D$), have a periodicity that the neural network does not directly understand. For instance, after 23h45 (timestep = 95) follows 00h00 (timestep = 0) and after Sunday (day = 6) follows Monday (day = 0). To overcome this issue, the initial features are transformed into cyclical values (sinus and cosines) using (11) for quarter-hours and (12) for day numbers [34].

$$\Delta T_{sin/cos} = \begin{cases} sin(\Delta T \times 2\pi \times \frac{1}{96}) \\ cos(\Delta T \times 2\pi \times \frac{1}{96}) \end{cases} \tag{11}$$

$$D_{sin/cos} = \begin{cases} sin(D \times 2\pi \times \frac{1}{7}) \\ cos(D \times 2\pi \times \frac{1}{7}) \end{cases} \tag{12}$$

### 3.2.3. Final Configuration

The three different LSTM configurations are summarized in Table 2. Additional explanations on how the parameters of the table have been defined are detailed in the next paragraphs.

The batch size is a trade-off between training speed and accuracy. The smaller the batch size, the faster the convergence will be. However, larger batches can reach lower minima than smaller batches [35]. In addition, the batch size can also be chosen in function of the nature of the data. Since the optimization speed is not an issue in this research, it has been decided to set the batch size equal to two days data (96 timesteps times 2).

**Table 2.** Neural network characteristics.

| Parameters | LSTM-B | LSTM-C | LSTM-W |
|:---:|:---:|:---:|:---:|
| Epochs | 30 | 50 | 50 |
| Batch size | 512 | 192 | 192 |
| Optimizer | RMSprop | Adam | Adam |
| Loss function | MAE | MSE | MSE |
| Learning rate | 0.001 | 0.001 * | 0.001 * |
| Hidden neurons | 16 | 25 ** | 30 ** |
| Activation function | Tanh | Tanh ** | Tanh ** |
| Dropout | 0.3 | 0 ** | 0.3 ** |

(*) learning rate is variable as given by (13) and (**) hyperparameters.

Configurations LSTM-C and LSTM-W use the *Adam* optimizer, similarly to [24], because it combines the advantages of the *RMSprop* optimizer and the *AdaGrad* optimizer [36]. The optimizer is evaluated using a mean-square error (MSE) loss function since the forecast results performed better than when using the mean-absolute error (MAE) loss function.

It has been decided to add a variable learning rate (*) in function of the number of epochs. The goal is to avoid either having a too slow convergence (small learning rate) or to avoid training divergence (too big learning rate) [37]. The variable learning rate is given by (13). Thanks to this, performances are slightly enhanced.

$$\text{learing rate} = \begin{cases} 0.001 & \text{when} \quad epochs < 20 \\ 0.0005 & \text{when} \quad epochs \geq 20 \end{cases} \tag{13}$$

In a neural network, some parameters cannot be learned by the network itself and have to be defined manually before the actual training. These parameters, called hyperparameters, need to be adjusted manually by either trial-and-error or using a hyperparameter optimizer. The optimizer used in this paper is a Bayesian optimization that uses Gaussian processes, implemented in Python using *Scikit-optimize* library. The parameters indicated with a double asterisk (**) in Table 2 have been hyperparameterized.

### 3.3. Forecast Post-Processing

To enhance the forecast of the neural networks, additional post-processing is performed based on observations done on the data used in this paper. The first rule applied to the forecast consists of replacing any negative power forecast to zero, since it is impossible to have negative aggregated EV charging demand. The second rule applied to the forecast consists of replacing the night aggregated EV charging demand to zero as well. This can be justified thanks to the data analysis performed in Section 4.1 where it is shown that between midnight and 5 am, the mean aggregated EV charging demand is zero. Both rules have been applied directly on the forecast and have a positive small impact.

### 3.4. Model Performance Evaluation

Two different well-known evaluation metrics are computed to assess the accuracy of the forecast. The first metric, called root-mean-squared error (RMSE), is given by (14):

$$RMSE = \sqrt{\frac{1}{N} \sum_{t=1}^{N} (\hat{P}_t - P_t)^2} \tag{14}$$

where $N$ is the number of timesteps (e.g., $N = 96$ timesteps for day-ahead forecast), $\hat{P}_t$ is the power forecast at timestep $t$, and $P_t$ is the real power at timestep $t$. The second metric, called mean-absolute error (MAE), is given by (15).

$$MAE = \frac{1}{N} \sum_{t=1}^{N} |\hat{P}_t - P_t| \tag{15}$$

## 4. Results and Discussions

### 4.1. Use Case and Data Analysis

The charging sessions have been recorded in a hospital semi-public charging site which consists of six chargers containing two Type 2 connectors of 22 kW each. The charging sessions have been recorded from mid-June 2018 until end-July 2019. Most of the charging sessions (95.2%) are from commuters, charging frequently at the chargers. Nevertheless, the EV charging sessions follow a constant pattern with a high stochastic behavior, as shown in Figure 4, where quarter-hours of the year are summarized in mean values, first and third quartile values, and the maximum values. The power profile is characterized by a typical working behavior where commuters arrive in the morning and leave in the afternoon. The maximum values indicate the high variability in average charging power where maximum values can go up to 30 kW (including charging flexibility) which is six times higher than the mean value.

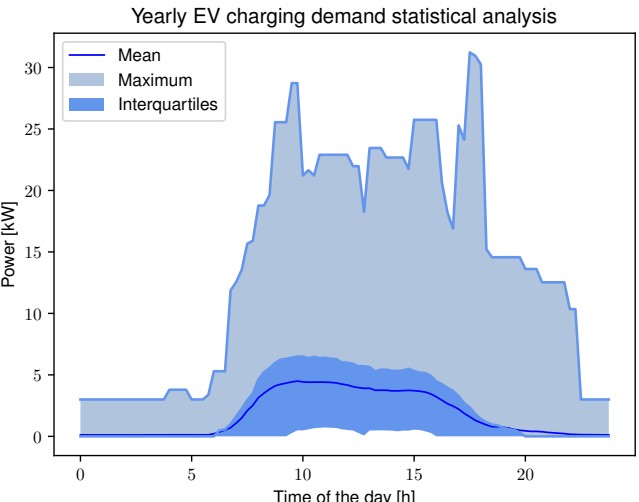

**Figure 4.** Yearly EV charging demand summarized to one day time where the mean, quartiles, and maximum values are shown.

Additionally, the weekly average EV charging demand is shown in Figure 5. The figure illustrates that there are weeks with higher average EV charging demand and weeks with lower average EV charging demand. For instance, weeks 28 and 29 represent the period between Christmas and new year vacations. Consequently, the weekly variability is included in the input dataset, so that the neural networks are informed of higher or lower EV charging demand. In addition, Figure 5 shows the subdivision of the dataset into training, validation, and test subsets.

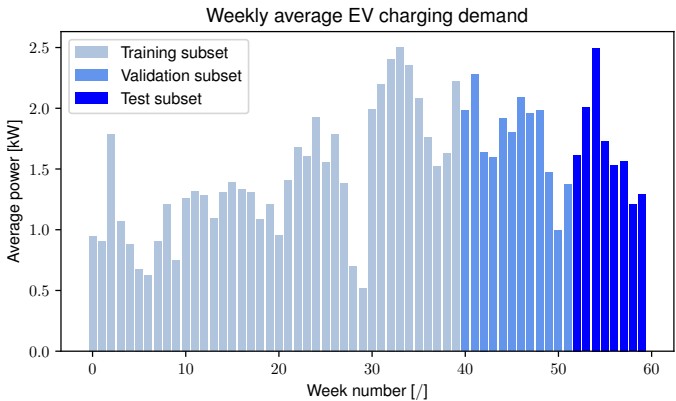

**Figure 5.** Weekly EV charging demand. Different colors are given for the training, validation, and test dataset.

### 4.2. Simulation Results

#### 4.2.1. Neural Network Convergence Analysis

For each neural network, two important curves are presented in Figure 6. These curves indicate if the training of the neural networks correctly convergences to a minimum MSE.

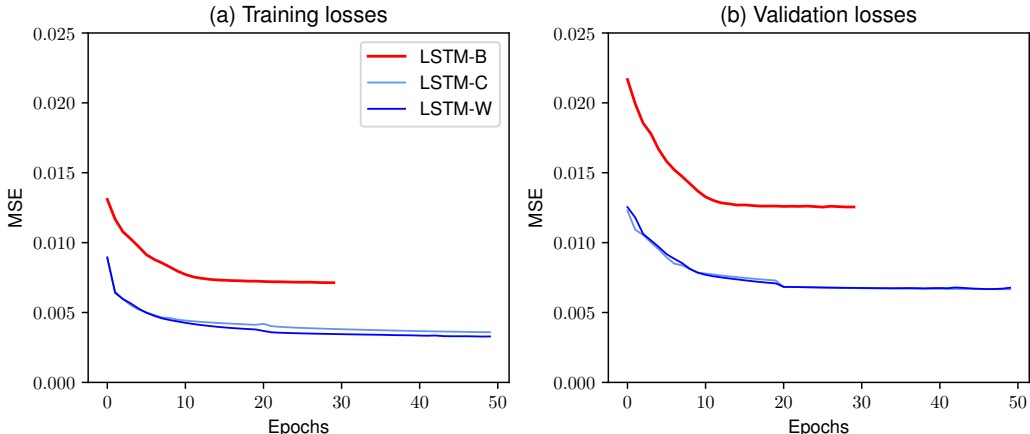

**Figure 6.** Training and validation losses expressed in mean-square error (MSE).

Firstly, an important difference between the minimum MSE of LSTM-B and of the two new neural networks can be observed. This result shows that LSTM-C and LSTM-W perform better by converging to a lower MSE than LSTM-B. Secondly, the difference between LSTM-C and LSTM-W is almost negligible. Nevertheless, these curves show that the three neural networks correctly converge for both training and validation losses and that there are neither over-fitting nor oscillation issues.

#### 4.2.2. Two-Weeks Period Forecast Example

A two-weeks period time taken from the test subset which consists of 14 day-ahead forecast examples, generated each time at midnight, is presented in Figure 7. The figure shows the real EV charging demand and the three different neural networks' forecasts. To have a better quantification, the RMSE and the MAE are computed and shown in Figure 8 for each day of the two-weeks period time.

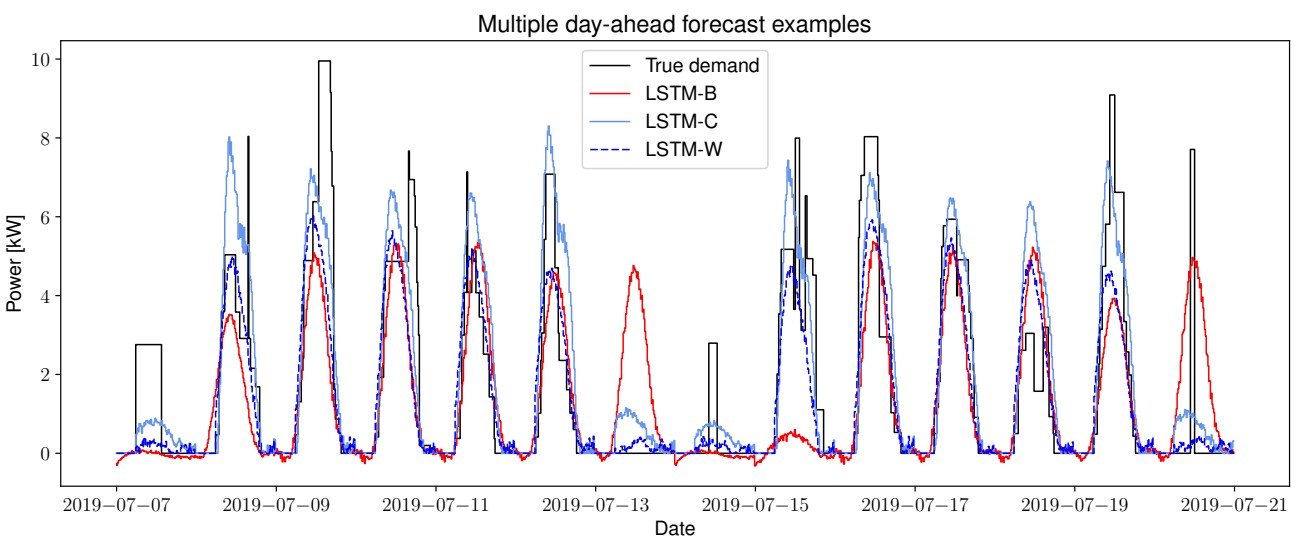

**Figure 7.** 14 day-ahead forecast examples using three different neural networks.

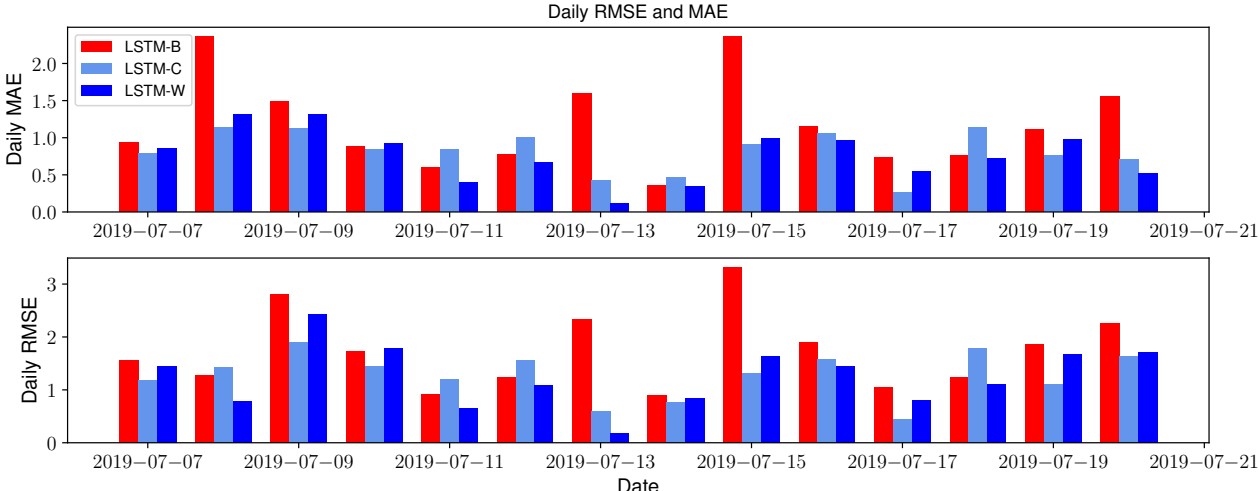

**Figure 8.** Daily RMSE and MAE of the two week forecast example.

The first and main observation is the excellent performances of LSTM-C and LSTM-W compared to LSTM-B thanks to additional calendar features. For instance, this can be observed on the 13th of July (Saturday) where LSTM-B forecasts a working day, whereas the two new neural networks successfully forecast the Saturday. Another example can be observed on the 15th of July. For these two specific days, LSTM-W performed best with a reduction in RMSE by up to 92.7% and with a reduction in MAE by up to 92.5%.

A second observation is the difficulty of having a satisfactory forecast during weekend days. Since EV users that charge during the weekend appear to have a random behavior (mainly non-commuters), the neural networks have difficulties to accurately forecast. For instance, on the 7th of July and on the 14th of July, one charging session occurred that was not predicted by any of the neural networks. Still the RMSE and MAE of LSTM-C and LSTM-W are low for weekends, as indicated in Figure 8 with hatched lines.

Regarding working days, both LSTM-B and LSTM-C understand the difference between different working days. For instance, on the 12th of July, Figure 7 shows that the LSTM-C is able to predict the peak demand. However, in general, a drawback of the method to construct the EV charging demand (in Section 3.1) is reflected by discrete steps in the real aggregated EV charging demand that the neural networks are not able to accurately predict.

### 4.2.3. Test Subset Performances

The average MAE and RMSE for the full test subset and for each neural network, including the relative reduction in error compared to LSTM-B, are shown in Table 3.

**Table 3.** MAE and RMSE results for each individual neural network.

| Metrics | LSTM-B | LSTM-C | LSTM-W |
|---------|--------|--------|--------|
| MAE | 1.25 kW | 0.96 kW (−23.2%) | 0.89 kW (−28.8%) |
| RMSE | 2.29 kW | 1.85 kW (−19.22%) | 1.92 kW (−16.16%) |

On a MAE perspective, LSTM-W performed the best, followed by LSTM-C and then LSTM-B. On a RMSE perspective, LSTM-C outperformed LSTM-W, followed by LSTM-B. This means that LSTM-C performs better in extreme cases (peak power), which can also be observed in Figure 7. In conclusion, adding calendar features increases the performance of the forecaster in extreme cases (peaks), whereas adding weather features increases the performances on MAE perspective.

#### 4.2.4. Feature Importance

The layers of the neural networks presented in this paper are interconnected by weights which need to be defined during the training of the neural networks. In particular, weights connected to the input layer can be analyzed in order to have an understanding on how the neural networks interact with the features. To do so, the algorithm of [38] has been applied on LSTM-C and LSTM-W to analyze the importance of the features. They calculate the importance of the features based on a simple underlying principle of allocating a higher importance to bigger weights and weights which are more susceptible to variations during training. Their algorithm is called Variance-based feature Importance in Artificial Neural Networks (VIANN) and is available in [39]. The results of such algorithm applied to both neural networks are shown in Figure 9.

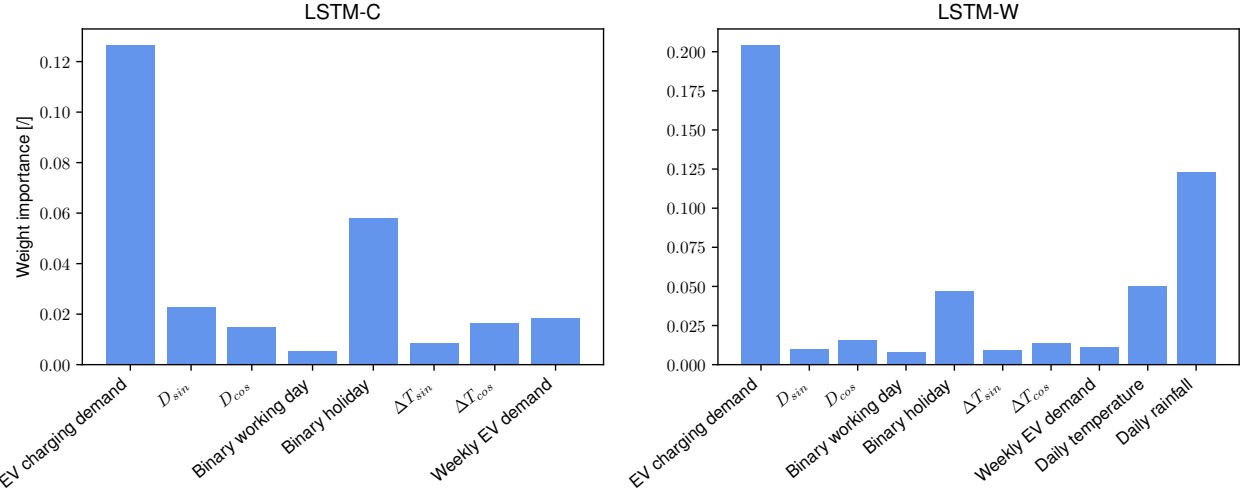

**Figure 9.** Features importance assessment.

The aggregated EV charging demand of the past 24 h feature has an important weight for both neural networks. This was expected since it is a inherent feature required by recurrent neural networks. For LSTM-C, the neural network allocates an important weight to the binary holiday feature. Regarding LSTM-W, the two additional weather features have also an important weight in the neural networks. The other LSTM-C and LSTM-W features have, to a greater or lesser extent, the same weight for the neural networks.

It is important to note that this feature-importance analysis does not necessarily rank the features based on their overall impact on the forecast results (RMSE and MAE of the test subset) because it does not take into account how often a feature manifests. For instance, the holiday feature is indicated to have a big importance yet has little effect on the overall MAE and RMSE because it does not influence the forecasting accuracy of any other regular day. This is confirmed using the intuitive Leave-One-Feature-Out (LOFO) analysis [40]. When a holiday manifests itself in the test subset, the results show that MAE and RMSE are respectively 5.78 and 6.11 times higher that day (15th of August—National holiday in Belgium) when removing the binary holiday feature.

This analysis provides an insight into the neural network dynamics and triggers further research into understanding the underlying behavior of features with respect to the forecast variable. The results highlight the fact that the binary holiday and weather features demonstrate differences in pattern when those features manifest and should thus be included to avoid large errors on those moments, even if there overall MAE and RMSE improvements are marginal.

#### 4.3. Forecast Timing and Real-Time Capabilities

A requirement for the forecasters in this paper is to be able to use them in real applications. This means that the forecasters should work in a sufficiently fast way, in

order, for instance, to use them (near) real-time. Accordingly, an analysis is performed on the time for training, validation, and testing of the neural networks. Such activities are done on an Intel® Xeon® E-2176M processor with 64 Gb of installed RAM. Table 4 shows the time to train together with the validation of the neural networks and the time to test 50 day-ahead forecasts with the average time to test one day-ahead forecast.

**Table 4.** Timing of training, validation, and testing of the neural networks.

| Neural Networks | Training and Validation [min] | Testing 50 Days [s] | Average Time for a Day-Ahead Forecast [s] |
|---|---|---|---|
| LSTM-B | 3.88 | 3.17 | 0.063 |
| LSTM-C | 11.47 | 3.42 | 0.068 |
| LSTM-W | 15.04 | 3.28 | 0.065 |

The results show a very small forecasting time of less than 0.1 s, which results in the possibility to use it in real-time applications. However, real-time application is only possible if the neural networks are trained beforehand using historical data. It can also be observed that the training and validation times of LSTM-C and LSTM-W are much higher than LSTM-B, mainly due to more data (more features) that have to pass through the neural network when trained.

## 5. Conclusions

The objective of this paper was to forecast the electric vehicle (EV) charging demand on a day-ahead horizon and on a 15 min time resolution. Two new forecast algorithms are presented and compared with a state-of-art algorithm extracted from the literature. The three algorithms are based on long-short-term memory (LSTM) neural networks. The algorithms differ mainly on the input data where the two newly presented algorithms include additional features such as calendar and weather features, in order to help the neural networks to capture the variabilities in daily EV charging demand. The neural networks are tested on a difficult use case of a hospital charging site. The results show great performance, despite the high variability and the high stochastic behavior of the EV charging demand pattern with a MAE lower than 1 kW. The root-mean-square error is reduced by 16.16% to 19.22% and the mean-absolute error (MAE) is reduced by 23.2% to 28.8% compared to the algorithm from literature (LSTM-B).

A feature importance analysis was conducted to understand the neural network dynamics and highlighted the relative importance of the holidays and weather features for the LSTM-C and LSTM-W forecasts, respectively. Future research could be performed in order to test the algorithms on different use cases and evaluate whether the underlying behavioral patterns manifest across the use cases. This will lead to the correct choice of appropriate forecaster for the chosen application according to their performance.

The day-ahead forecasters in this paper demonstrate an advancement to the state-of-the art with regards to accuracy. They are real-time capable with respect to the focus application of energy management. The forecasters allow for the enhancement of the optimization of the scheduling of electric vehicle charging by including the future charging demand.

**Author Contributions:** Conceptualization, C.D.C. and G.V.K.; methodology, G.V.K.; software, G.V.K.; validation, G.V.K., C.D.C. and N.S.; formal analysis, G.V.K.; data curation, C.D.C.; writing—original draft preparation, G.V.K.; writing—review and editing, C.D.C. and N.S.; visualization, G.V.K.; supervision, C.D.C. and N.S.; project administration, T.C. and M.M.; funding acquisition, T.C. and M.M. All authors have read and agreed to the published version of the manuscript.

**Funding:** This research was funded by "Agency for Innovation and Entrepreneurship (VLAIO)" grant number HBC.2018.0519.

**Data Availability Statement:** The datasets used during the current study are not available since they are protected by privacy reasons, mainly because of the presence of EV users' radio frequency identification (RFID).

**Acknowledgments:** The authors would like to thank Flux50 for support to our team.

**Conflicts of Interest:** The authors declare no conflict of interest.

## Abbreviations

The following abbreviations are used in this manuscript:

| | |
|---|---|
| EMS | Energy management system |
| EV | Electric vehicle |
| MAE | Mean-absolute error |
| MSE | Mean-square error |
| LSTM | Long short-term memory |
| LSTM-B | Long short-term memory-Base |
| LSTM-C | Long short-term memory-Calendar |
| LSTM-W | Long short-term memory-Weather |
| Relu | Rectified linear unit |
| RMSE | Root-mean-square error |
| RNN | Recurrent neural network |
| Tanh | Hyperbolic tangent |
| VIANN | Variance-based feature Importance in Artificial Neural Networks |

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
