# Peer review of "Day-Ahead Forecast of Electric Vehicle Charging Demand with Deep Neural Networks"

_wevj, doi:10.3390/wevj12040178_

Round 1

Reviewer 1 Report

1) The importance of considering calendar & weather should be described properly in the beginning of manuscript. Calendar is understandable as work/ off days affect the presence of EVs on the premises. Regarding weather, it is not so clear.

2) References should be made diverse. Multiple references from one group of authors is not a good practice. The authors have themselves mentioned that RNN/ LSTMs have been intensively used in recent research, despite that they cite a few set of authors repeatedly. Ref. [5,6] , [8,12,13], [9,10,11], [20,21]

3) Usage of eqs. 3 & 4 is interesting. Very good!

4) LSTM is not exactly a hidden layer in RNN (line 137). It is a type of RNN architecture. If authors are using LSTM, the notations should be replaced everywhere and the reason for using LSTM must be highlighted.

5) Regarding data pre-processing, I am curious how the number of EVs on the premises influences the demand. What do authors think about no. of EVs as a parameter of interest? 

6) Although the z-score normalization is more frequently used, the min-max normalization, after a preliminary test, showed clearly better performances - in terms of statisitical theory, why do you think min-max normalization suits better in some cases and in some cases z-score? Please explain. It is fine that the results show one is better than the other, but we must know why does that happen, in order to wonder - will it always be the case - if I add an extra variable or tweak the parameters? 

7) Section 2.2 is nice in terms of concise description of the process. To improve it further, authors must add graphs of parameters listed in Table 2. Graphs should show how the optimal parameters were identified over the epochs!

8) Authors used scikit for hyperparameter tuning. Which library did they use to develop the RNN/ LSTM? It would be nice to mention 2-3 lines about that as well. Maybe also mention which library did you use for figure 2. Normally, boxplots are used, but it is interesting to see the information shown in 24 hr format. Good!

9) What is the difference between epochs, iterations, and batch size? How does batch size influence LSTM performance? Why is it normally chosen in powers of 2?

10) Authors have completely missed out on mathemtical description of the LSTM/RNN. Since the main contribution of paper is application. Mathematical description is necessary. This is the weakest point of the manuscript. No equations, no schematics, no description of modeling.

11) There is an error in representation in Fig. 4. MAE or MSE? Correct it please.

12) Learning rate is very small in eq. (5). It would be nice to see the computational requirements. Please report the times taken for training, validation & testing in a table. 

13) Does the proposed forecasting algorithm work in real-time? Or the analysis was done a posteriori?

14) Why vairance based feature importance was deemed suitable by the authors? - shouldnt importance based feature selection be done before forecasting? 

15) D/T sin/cos based features don't seem to help much. What are your comments regarding that?

16) In Fig 6 both cases, RNN-W doesnt seem to offer much improvement on RNN-C. Which brings me back to my initial question - why did authors think weather data can really impact EV charging demand forecasting? Comments?

17) Last question - what is the value derived from this work? I can estimate power demand, how can I put it to use further - future directions?

Improvements definitely needed, but good work overall!

Reviewer 2 Report

The paper entitled "Day-ahead Forecast of Electric Vehicle Charging Demand with a Recurrent Neural Network"  proposes the enhancement of a state-of-the-art deep neural network to forecast the day-ahead charging demand of electric vehicles (EV) with a time resolution of 15 minutes and adding the new features to the neural network in order to improve the forecasting. Therefore, the paper tackles an important and timely issue that is marked by the widespread deployment of EVs as a part of decarbonization of the world's economy and all the issues it creates. The paper is technical rather than theoretical and needs better description of the methodology, longer literature review and elaboration of results. Please see my comments and suggestions below:

1) The paper lacks the literature review. More sources and references on this topic (and on EV charging in general) should be added. The list of references can be extended by 20-30 additional sources.

2) The methodology should be better explained and links to the previous studies need to be provided (see point 1). The examples from California, Norway, UK, Germany and other countries with a high share of EVs should be analyzed and compared.

3) The results and conclusions section needs to be written up. What are the implications for the EV owners or stakeholders? What lessons cane be drawn from this paper? What are the policy implications and pathways for further research?

4) The paper might benefit from minor English proofreading.

Reviewer 3 Report

This is an interesting study and a timely topic for a research work, focusing on day-ahead forecast of Electric Vehicle charging demand with a recurrent neural network. Although the authors have demonstrated great efforts with such an important study, the literature review should be further extended as part of the Introduction section in the manuscript. Otherwise, the paper would be very interesting to the readers working in the related field, and the conclusions derived from the study are also offering some very useful findings. I do not have any further comments or suggestions for further improvement of this paper.

Round 2

Reviewer 1 Report

-

Author Response

We would like to thank the reviewer.

Reviewer 2 Report

The paper has been modified but it still required extending the Literature review (which is now a separate sub-section). The current number of sources is still very low and some additional 20-30 sources (or more) would be beneficial. In addition, the Conclusions need to be written up and the policy implications and the pathways for further research belong to the Conclusions section and not to the discussion within the text.
